# *Agrobacterium fabrum* (*tumefaciens*) Chemosensory System: A Typical Model of One Histidine Kinase for Two Coupling Proteins and Multiple Response Regulators

**DOI:** 10.3390/microorganisms13112556

**Published:** 2025-11-09

**Authors:** Jinjing Liu, Mengya Feng, Nan Xu, Hao Wang, Minliang Guo

**Affiliations:** College of Bioscience and Biotechnology, Yangzhou University, Yangzhou 225009, China; DX120210194@stu.yzu.edu.cn (J.L.); MX120241110@stu.yzu.edu.cn (M.F.); 006580@yzu.edu.cn (N.X.); wanghao@yzu.edu.cn (H.W.)

**Keywords:** chemotaxis, chemoreceptor, signal transduction, chemotaxis protein, chemosensory complex, signal cross-talk

## Abstract

Bacteria utilize chemotaxis to sense the surrounding chemical signals to seek a more favorable survival environment. The chemotaxis process includes signal sensing, signal transduction, and signal response (i.e., regulating flagellar rotation to control motility). *Agrobacterium fabrum*, as a soil-born facultative phytopathogen, can survive in diverse environments from bulk soil, the rhizosphere, to the plant-associated niches, and needs to cope with diverse challenges from various survival environments. It must recognize a variety of environmental signals and thus has evolved a chemosensory signaling system more complicated than the prototypical chemotaxis system. The chemosensory system of *A. fabrum* possesses one histidine kinase, but more chemoreceptors, coupling proteins (2 CheWs), and response regulators (2 CheYs and 1 CheB) than the well-studied prototypical system in the model bacterium *Escherichia coli*, which has only one CheW, one CheY, and fewer chemoreceptors. More response regulators imply that the chemosensory system may involve other physiological functions beyond chemotaxis. In this review, we outline the recent advances in the prototypical chemosensory signaling system and discuss the functions of protein components in *A. fabrum*’s chemosensory system by comparing those proteins with the homologous proteins observed in the paradigm and other closely related species. Meanwhile, we place particular emphasis on reviewing the data about the chemosensory system of *A. fabrum*, propose a “one-system two-pathways” model depicting that *A. fabrum* possibly utilizes one histidine kinase to assemble two chemosensory signaling pathways, and envision future directions for studying this system. The insights provided will aid in understanding the diversity of chemosensory signaling pathways in other organisms and the molecular mechanism mediating the signal crosstalk among chemosensory signaling pathways.

## 1. Introduction

Bacterial chemotaxis is defined as the behavior of motile bacteria and archaea in which they track a variety of environmental cues and move toward more favorable environment [1]. Through sensing the environmental attractants or repellents, bacterial cells can efficiently locate optimal growth environments [2,3,4]. Bacterial chemotaxis utilizes a highly conserved chemosensory signaling system to transduce signals. Chemotaxis is essential not only for bacteria to locate more favorable environments under nutrient stress [5], but also for various human [6], animal [7,8], and plant pathogens to invade their hosts [9,10,11]. Chemotaxis also significantly affects biofilm formation [12,13,14], adhesion ability [15], polymicrobial communication [16,17,18], bacterial symbiosis with plants [19,20,21,22], and the regulation of second messenger levels [23,24,25]. In addition, chemotaxis is very important in enabling bacteria to move toward biodegradable pollutants, and thus plays a crucial role in the biodegradation of some pollutants that may pose serious environmental and health risks [2,26].

Molecular study of bacterial chemotaxis began in the 1960s, and was pioneered by Adler’s work [1,27]. For over six decades, the molecular mechanism of bacterial chemotaxis has been well understood. The chemosensory pathway begins with the sensing of chemical gradients through chemoreceptors, and then the sensory signals are processed by the core signaling unit (CSU) and transmitted to the flagellar motor by cytoplasmic proteins, causing the bacteria to “run” or “tumble” via the change in flagellar rotation. The chemosensory pathway used by chemotaxis is highly conserved and has been identified in more than half of the sequenced bacterial genomes [28,29]. The chemosensory pathway represents a major signal transduction mechanism in bacteria and has become a paradigmatic model for the study of bacterial signal transduction [30]. What is more important is that the chemosensory system exhibits significant diversity in terms of the proportional composition of core proteins and the involvement of auxiliary proteins [31]. Moreover, the phenomenon of multiple chemosensory systems occurring in a single bacterium is also very common [29,32,33].

*Agrobacterium fabrum* (formerly known as *Agrobacterium tumefaciens*) is a remarkable and well-studied bacterium that has significantly influenced the fields of plant biology and agricultural biotechnology [34,35]. This Gram-negative soil bacterium is renowned for its unique ability to transfer the segments of its DNA, known as the T-DNA (transferred DNA), into the cells of plants [36]. In nature, the transfer and integration of the T-DNA into the plant host genome leads to the production of abnormal plant growths, or tumors, which is how *Agrobacterium fabrum* earned its former name “*tumefaciens*” (meaning tumor-forming) [20,37]. However, this ability to transfer DNA has also made it an invaluable tool in genetic engineering to create transgenic plants with improved traits [38,39]. *A. fabrum* is a facultative phytopathogen and able to live in diverse environments, such as soil, the rhizosphere, and the plant-associated niches, which are challenged by multiple stressors, such as diverse nutritional sources, microbial competition, and plant defenses [40,41]. It is required to sense various environmental signals. In fact, *A. fabrum* has chemotaxis responses to a variety of chemical substances [2,42]. Over twenty genome sequences derived from *A. fabrum* are publicly available [40]. All the published genomes of *A. fabrum* strains are annotated to encode a single complete set of core chemotaxis proteins—without the presence of multiple sets or variations in the component proportions of core proteins among different strains [43], demonstrating that neither multiple sets nor compositional variations in core chemotaxis proteins occur in different *A. fabrum* strains, though the number of chemoreceptors may differ among strains from different ecological environments. Such composition of core chemotaxis proteins is also found in other plant-associated bacteria, such as *Sinorhizobium meliloti* [44,45], *Azorhizobium caulinodans* [46], *Rhodobacter sphaeroides*, and *Caulobacter crescentus* [47]. Therefore, *A. fabrum*’s chemosensory system may potentially represent a distinct class of chemosensory systems prevalent in plant–microbe symbionts. A review of this chemosensory system will be helpful for understanding the diversity of chemosensory pathways and the ecological functions of the chemosensory system in plant-associated bacteria. Notably, while numerous reviews have addressed prototypical chemosensory systems, none have dealt with the recent advances in *A. fabrum*’s chemosensory system.

## 2. Prototype of Chemosensory Signaling System and Co-Existence of Multiple Systems

Most bacterial signaling transduction adopts a two-component system, which consists of a sensory kinase and a response regulator. Typically, the sensory kinase is a transmembrane protein that senses the extracellular signal and transmits the signal to the cytoplasmic response regulator through phosphorylating the latter [2]. The phosphorylated response regulator regulates the corresponding physiological functions [48]. The chemotaxis signaling pathway is a special case of a two-component system [2,31]. The sensory kinase of the chemotaxis pathway is divided into three different proteins: chemoreceptor (or methyl-accepting chemotaxis protein, MCP), coupling (or adaptor) protein, and histidine kinase. Such functional segregation of signal sensing (chemoreceptor) and phosphorylation activation (histidine kinase) might be beneficial to the evolutionary adaptation of the chemotaxis system to sense different chemoeffectors in new environmental niches via the acquisition or loss of specific chemoreceptors without affecting the function of the histidine kinase [49,50].

Besides chemotaxis, this chemosensory signaling system is reported to be involved in other physiological functions [8,11]. According to their functions, chemosensory signaling systems can be classified into three groups: those regulating the flagellar motility (Fla), type IV pilus-based motility (Tfp), and those involved in non-motility functions (or alternative cellular functions, ACF) (such as biofilm formation, cell morphology, and the control of second messengers) [11,31]. The Fla group is the most diverse and further classed into 17 subclasses (F1–F17), while Tfp and ACF groups contain a single system class [11,29,31]. The core of the chemosensory signaling system contains at least four essential components, namely the chemoreceptor (MCP), coupling protein (CheW), histidine kinase (CheA), and response regulator (CheY).

### 2.1. Prototype of Chemosensory Signaling System

The signaling pathway of the chemosensory system is highly conserved across prokaryotes and best understood in *Escherichia coli* [28,31]. The chemosensory signaling system consists of two signaling modules: one for rapid signal transduction to regulate the physiological functions and another for slower adaptation to the new environmental signal strength [49]. The rapid signal transduction module comprises the MCP–CheW–CheA ternary complex (Figure 1A) and CheY. The binding of ligand to the chemoreceptor MCP causes the conformation change that is transmitted to the histidine kinase CheA and modulates the kinase activity. The kinase CheA can phosphorylate the response regulator CheY. Therefore, CheA, together with a phosphatase CheZ that can specifically dephosphorylate the phosphorylated CheY (CheY-P), controls the cytoplasmic level of CheY-P. The diffusible CheY-P regulates the corresponding functions. For chemotaxis, CheY-P binds to the flagellar motor to change the flagellar rotation. The slower adaptation module is achieved by a methyltransferase CheR and a methylesterase CheB [51]. CheR is an MCP-specific methyltransferase that can methylate the glutamyl residues of chemoreceptors. Due to its ability to accept methyl groups, the chemoreceptor is named as the methyl-accepting chemotaxis protein (MCP). CheB can also be phosphorylated by CheA. The phosphorylated CheB (CheB-P) removes methyl groups from the MCPs and controls the methylation state of the MCPs in conjunction with CheR. The methylation state of the MCPs affects the MCPs’ ability to activate CheA and thus adjusts MCPs’ sensitivity to the ligand to adapt the new ligand concentration [52,53].

Typical chemoreceptors, which sense the environmental signal outside the bacterial cell, are generally transmembrane proteins containing a periplasmic ligand-binding domain (LBD), a transmembrane (TM) helical region, and a cytoplasmic signaling domain (CSD), which comprise four regions (or sub-domains), the HAMP (found in histidine kinase, adenylyl cyclase, methyl-binding protein, and phosphatase) domain, methylation helix (MH) bundle, flexible region (or glycine hinge, GH), and protein-interacting region (or kinase control region, KC) [54]. Two chemoreceptors form a homodimer. Three chemoreceptor dimers further make up a trimer-of-dimers. Two trimers-of-dimers, together with one CheA dimer and two CheW monomers, constitute the smallest functional unit, called the core signaling unit (CSU) (Figure 1A). CheA and CheW bind to the kinase control region of the chemoreceptor to form the baseplate of the CSU (Figure 1B,C) [51,55,56]. Many CSUs then form into larger structures, in which the trimers-of-dimers of the chemoreceptors are packed into a supramolecular array across the bacterial inner membrane at the cell pole. The interaction between the CheW and CheA P5-domain forms the alternating hexameric rings that link the CSUs together to assemble the baseplate of the supramolecular array. Variation in the ratio of CheA to CheW can change the composition of the hexameric ring. Additional CheW molecules can form hexameric CheW rings in the supramolecular array (Figure 1D). Recent results have demonstrated that the hexameric CheW rings enhance the response sensitivity and cooperativity of the chemosensory array, but are not indispensable for the assembly of CheA-containing arrays [57].

In recent years, studies on the detailed structure of the CSU and the signaling mechanism by which the transmembrane chemoreceptor propagates the periplasmic ligand-binding signal to its membrane-distal tip, which is 200 Å away, to control the kinase CheA have made great strides. The sub-nanometer resolution three-dimensional map of a complete CSU was acquired by using cryo-electron tomography, which reveals the structure of the CSU in unprecedented detail [30,58,59,60]. More recently, the combination of cryo-electron tomography and molecular simulation presents the complete atomic structure of the native CSU, which provides new insight into the interactions between neighboring chemoreceptor ligand-binding domains, and elucidates previously unresolved interactions between individual CheA domains [56]. Hydrogen deuterium exchange mass spectrometry was employed to measure the signaling-related allosteric changes in the chemoreceptors, CheW and CheA within functional complexes, revealing that the cytoplasmic domain of the chemoreceptor remains disordered even within functional complexes and this intrinsic disorder plays a crucial mechanistic role in controlling CheA kinase activity through long-range allosterically stabilizing the catalytic domain of CheA [61,62]. Protein in vivo cross-linking and Förster resonance energy transfer (FRET)-based kinase assays were used to distinguish the transitional changes in each subunit in the structure and interacting interface, which are caused by various dynamic stimulus signals [63,64,65]. All these recent results are very useful for understanding the details of how the signals are mechanically transmitted to CheA.

### 2.2. Co-Existence of Multiple Chemosensory Systems

Comparative analysis of bacterial genome sequences demonstrated that the chemosensory system presents in most prokaryotic species and that more than half of all motile bacteria have multiple chemosensory signaling systems in their genomes [29,66]. Experimental evidence from both genetic investigation and cryo-EM observation also demonstrated that bacteria may express multiple chemoreceptor arrays segregated into distinguishable assemblies [11,67]. An “average” bacterial genome has 14 chemoreceptor genes [68]. The cytoplasmic domains of various chemoreceptors have different lengths due to the number of seven-residue repeats inserted in the cytoplasmic signaling domain although they are highly conserved [69]. Arrays are only formed among chemoreceptors with the same physical length, resulting in distinct chemoreceptor arrays in many bacterial cells [70,71]. Recently, more and more experimental observations have demonstrated that different types of chemoreceptor arrays present in a single bacterial cell [32,72,73,74,75]. Moreover, protein components encoded by the genes of different chemotaxis system clusters can assemble into promiscuous chemosensory signaling arrays [76,77]. The signals transduced by various chemosensory signaling arrays may be involved in different physiological functions. The coexistence of multiple chemosensory signaling pathways within a cell results in the crosstalk between chemosensory systems [13,72,76] and raises the questions of how these signals from different systems are coordinated in the signal transduction and functionally separated at the level of response regulators [78].

## 3. Chemosensory-Related Proteins Encoded by *Agrobacterium fabrum* Genome

Comparative genomics analysis of 22 *A. fabrum* genomes showed that all *A. fabrum* genomes encode only one chemosensory histidine kinase CheA, two CheWs, and two CheYs, signifying that *A. fabrum* has only one complete set of core chemosensory protein components and that the chemosensory systems in different strains are highly similar, although different *A. fabrum* genomes encode different numbers of chemoreceptors [43]. In this review, we will only discuss the chemosensory system in the model strain C58 of *A. fabrum* due to the differences in some accessory chemosensory proteins and chemoreceptors in different strains. The genome of *A. fabrum* C58 carries one chemotaxis gene cluster, which encodes most of the main chemosensory protein components, including one copy of CheA, CheB, CheR, CheD, CheS, and MCP and two copies of CheY. Besides the chemotaxis gene cluster, *A. fabrum* C58 has two CheW-encoding genes and an additional nineteen MCP-encoding genes scattered on different locations of the genome [20,43].

### 3.1. Chemoreceptor (Or Methyl-Accepting Chemotaxis Protein, MCP)

Bacterial chemoreceptors detect a variety of signals. In fact, they vary widely in sensing mode, cellular location, protein topology, and above all, the type of LBD, even though their CSDs are highly conserved. Over a hundred different types of LBD have been identified among chemoreceptors, but only a few of them are prevalent [79]. Based on LBD and membrane topology, chemoreceptors can be classified into four major classes: class I, the typical transmembrane chemoreceptor class with a periplasmic LBD; class II, the transmembrane chemoreceptors with an N-terminal cytoplasmic LBD; class III, the transmembrane chemoreceptors with a cytoplasmic LBD after the last transmembrane helix; and class IV, the cytoplasmic (soluble) chemoreceptors [80]. Some LBDs of the same type recognize structurally very different ligands, whereas some ligands can be recognized by a number of different LBD types [81]. The lack of a structure–function correlation in LBDs prevents a reliable prediction of the function of chemoreceptors by extrapolation from the experimentally studied homologs. Moreover, some chemoreceptors, especially those without LBDs, indirectly recognize their ligands through the interaction partner proteins [3,82].

*A. fabrum* C58 has a total of 20 chemoreceptor-encoding genes. Only one gene is located on the chemotaxis gene cluster, while 19 genes are scattered in various locations of the genome. In addition to 13 on the circular chromosome and 5 on the linear chromosome, the Ti and At plasmids each carry one chemoreceptor-encoding gene [20]. The LBDs and categories of all 20 chemoreceptors are summarized in Figure 2. Among the 20 chemoreceptors, 14 contain at least one transmembrane region: 12 are typical transmembrane chemoreceptors with periplasmic LBDs, and 2 are transmembrane chemoreceptors with cytosolic LBDs. The remaining six chemoreceptors lack transmembrane regions and are classified as cytosolic chemoreceptors [43]. Twelve chemoreceptors carry four known LBD types (calcium channels and chemotaxis, CACHE; cyclases/histidine kinases associated sensory extracellular fold, CHASE; found in Per-Arnt-Sim protein, PAS; Protoglobin), while the LBD types of the remaining eight are unknown [83]. The functions of seven *A. fabrum* chemoreceptors have been experimentally verified: (1) The *atu0514*-encoded chemoreceptor affects the chemotactic responses of *A. fabrum* to a broad range of chemoeffectors [84]; (2) The *atu0526*-encoded chemoreceptor is the only chemoreceptor that recognizes formic acid [85]; (3) The *atu1912*-encoded chemoreceptor senses pyruvate and propionate [86]; (4) Deletion of the *atu1027* gene abolishes the aerotactic response of *A. fabrum* to atmospheric air [87]; (5) The chemoreceptor encoded by *atu0373* is the primary chemoreceptor for choline, acetylcholine, betaine, and L-carnitine in *A. fabrum* [88]; (6) The *atu0872*-encoded chemoreceptor is a phenolic acid-sensing one [89]; and (7) The *atu2173*-encoded chemoreceptor broadly mediates amino acid chemotaxis, responding to all 17 proteinic amino acids except Glu, Asp, and Gly [90].

According to the classification of chemoreceptors based on the number of heptad repeats in the flexible region, all of the *A. fabrum* chemoreceptors belong to the category of 36H [70]. Sequence alignment of 20 *A. fabrum* chemoreceptors showed that their sequences in the protein interaction region, which interacts with CheA and CheW, are highly conserved with 95% identical residues. However, individual key residues, which are directly involved in the interaction with CheA and CheW, show divergence among 20 chemoreceptors, implying that different chemoreceptors possibly possess different affinities to interact with CheA and CheW. Sequence analysis of 20 *A. fabrum* chemoreceptors also showed that only 9 chemoreceptors possess a C-terminal pentapeptide for tethering two chemotaxis-adaptational proteins, CheB and CheR.

### 3.2. Coupling Protein, CheW

CheW is an adaptor protein that couples the histidine kinase CheA to chemoreceptors to form the CSU [91,92]. The genome of *A. fabrum* C58 carries two CheW-encoding genes, *atu2075* and *atu2617*. Neither is located on the chemotaxis gene cluster, but downstream of *atu2617* lies a cytosolic chemoreceptor-encoding gene, *atu2618* [93]. We are not sure whether this gene organization indicates that CheW encoded by the *atu2617* gene preferentially couples CheA to the cytoplasmic chemoreceptors rather than the transmembrane chemoreceptors. These two CheWs share 47% identical residues, but key residues involved in the interactions with the chemoreceptor and CheA show high divergence, suggesting that they may have distinct affinities for coupling CheA to different chemoreceptors. Reverse genetic experiments via gene deletion verified that CheW_2617_ (encoded by *atu2617*, CheW2) had a significantly more severe impact on *A. fabrum* chemotaxis than CheW_2075_ (encoded by *atu2075*, CheW1), even when both CheWs were expressed by using the same system and thus had identical molecular counts [93]. However, both CheWs are very important for the chemotaxis of *A. fabrum*. Only when both CheWs are deficient does *A. fabrum* chemotaxis completely disappear. Experimental results also showed that the molecular counts of the two CheWs differ in the wild-type cells [93].

### 3.3. Histidine Kinase, CheA

CheA is the key player in the chemosensory signaling system and responsible for transducing the allosterically mechanistic signal from the chemoreceptors into the chemical signal of phosphorylated CheY (CheY-P). In the prototypical CSU, CheA exists in dimeric form and one CheA subunit uses ATP to trans-phosphorylate the other. Phosphorylated CheA then transfers the phosphoryl group to the response regulators, such as CheY or CheB [2]. CheA proteins from *E. coli* and *Thermotoga maritima* consist of five domains. From the N-terminus to the C-terminus, the five domains of CheA are labeled P1 through P5 (P1: histidine-containing phosphotransfer domain, Hpt; P2: P2/CheY-binding domain; P3: dimerization domain, H-kinase_dim; P4: histidine kinase domain, HATPase_c; P5: CheW-like domain, homologous to the CheW protein) [94]. These five CheA domains were recognized as members of conserved domain families. However, comparative analysis of 13,673 CheA protein sequences from 7367 genomes showed that only 46% of the CheA protein sequences have the classical five domain architecture [95].

The genome of *A. fabrum* C58 encodes only one CheA. *A. fabrum* CheA is encoded by the *atu0517* gene located on the chemotaxis gene cluster. We compared *A. fabrum* CheA with *E. coli* CheA and found that *A. fabrum* CheA has the classical five domain architecture, but contains 98 additional residues inserted between the P2 and P3 domains as compared with *E. coli* CheA. It is unknown whether the long residue insertion is very important for the function of *A. fabrum* CheA. Moreover, only 30.83% of the residues of *A. fabrum* CheA are identical to those of *E. coli* CheA. In particular, when the corresponding domains of *A. fabrum* CheA and *E. coli* CheA are individually compared, the P2/CheY-binding domain shows the greatest sequence divergence, with all other domains exhibiting less divergence than this domain, which implies that the binding between CheA and CheY in *A. fabrum* may adopt a mechanism different from that in *E. coli*.

### 3.4. Response Regulators, CheY and CheB

Response regulators belong to a super protein family that regulates diverse signal relays via the phosphorylation of one of their conserved aspartate residues. They are characterized by the presence of a conserved receiver (REC, or Response_reg in Pfam) domain with a β5α5 fold structure, a central five-stranded parallel β sheet surrounded by five α helices [96]. The conserved aspartate residue in the receiver domain of the response regulator accepts the phosphoryl group from auto-phosphorylated histidine kinase, resulting in response regulator activation. In most cases, the response arising from the activated response regulator should be terminated in a timely manner. Therefore, all response regulators are able to auto-dephosphorylate. However, auto-dephosphorylation is often unregulated and thus, in some cases, bacteria have to evolve a specific phosphatase to remove the phosphoryl group from the activated response regulator [97]. Usually, a typical chemosensory system contains two types of response regulators, CheY and CheB.

In general, CheY is used to regulate the flagellar motor [98]. However, many bacteria have multiple CheYs and some CheYs do not regulate the flagellar motor [99]. For example, both *Azorhizobium caulinodans* [46,97] and *Rhizobium meliloti* [45] encode only one CheA, but two CheYs that have a clearly different influence on the chemotactic response. The chemotaxis gene cluster of *A. fabrum* C58 has two CheY-encoding genes, *atu0516* and *atu0520*. Both *A. fabrum* CheY variants possess the typical β5α5 fold structure, but they share only 35.66% identical amino acid residues and their residues corresponding to the *E. coli* CheY residues that were identified to interact with CheA and motor protein FliM are divergent. CheA-dependent cellular CheY localization assays and in vitro pull-down experiments showed that the affinity of *atu0516*-encoded CheY (CheY_516_, CheY1) to CheA is approximately 1.6-fold higher than that of *atu0520*-encoded CheY (CheY_520_, CheY2), but the affinity of CheY2 to FliM is approximately 5-fold higher than that of CheY1 [100]. These results suggest that two *A. fabrum* CheYs have distinct functions. The function of regulating flagellar motility is likely performed by CheY2, whereas CheY1 probably has additional physiological functions beyond flagellar motility regulation.

CheB is the other type of response regulator in the chemosensory system. Unlike CheY, which consists solely of a REC domain, CheB is a typical response regulator comprising two domains (an N-terminal REC domain and a C-terminal methylesterase/deamidase effector domain) connected by a long linker region [101,102]. In fact, over three-fourths of response regulators (approximately 77%) are two-domain proteins containing a REC domain and an effector (or output) domain. The REC domains in various response regulators are highly conserved, whereas the effector domains, which are responsible for outputting the response, vary widely in structure and function depending on the specific response regulator [96]. CheB’s effector domain is a methylesterase/deamidase enzyme domain that can catalyze the demethylation of specific methyl glutamate residues formed by methyltransferase CheR on the chemoreceptors and the deamidation of specific glutamine residues in the methyl-accepting region of the chemoreceptors to create some methyl-accepting sites [103]. However, some CheBs, for example, *Bacillus subtilis* CheB, may not have the deamidase activity. Similarly to CheY, CheB’s catalytic activity is activated through phosphorylation via CheA. Phosphorylated CheB (CheB-P) is tethered to the C-terminal pentapeptides of chemoreceptors and then recruited to the chemoreceptor substrate [33,104,105]. CheB is recruited to the chemoreceptor array as stimulated by a repellent and dissociates from the array once adaptation is complete [106].

*A. fabrum* CheB is encoded by the *atu0519* gene in the chemotaxis gene cluster and has two CheB-characterized domains. CheB is highly conserved. *A. fabrum* CheB has 46.76% of its residues identical to those of *E. coli* CheB, 46.91% of its residues identical to those of *Pectobacterium atrosepticum* CheB, and 47.61% of its residues identical to those of *Salmonella typhimurium* CheB. The crystal structures of both *P. atrosepticum* CheB and *S. typhimurium* CheB were solved. According to the comparison of the sequences and structures of these CheBs, the residues in the long linker between the two domains are more divergent than the residues in the REC and effector domains. However, the chemoreceptor-tethering region of CheB is located on the N-terminal part of the linker with high residue divergency. Therefore, the high residue divergency in this region results in the fact that CheBs from various bacteria have significantly different affinities to tether to the C-terminal pentapeptide of the chemoreceptor. For example, CheBs from *E. coli* and *S. typhimurium* can tether to the C-terminal pentapeptide of the chemoreceptor, but *P. atrosepticum* CheB cannot [107]. Nine *A. fabrum* chemoreceptors possess the C-terminal pentapeptide and thus *A. fabrum* CheB should have the capacity to recognize the C-terminal pentapeptide.

### 3.5. Methyltransferase and Deamidase for Chemoreceptor Modification, CheR and CheD

CheR is highly conserved and exists in almost all chemosensory systems [29]. *A. fabrum* CheR was annotated to be encoded by the *atu0518* gene. CheR is an S-adenosylmethionine-dependent methyltransferase, which belongs to a subclass of the super methyltransferase family that catalyzes the methylation of a variety of substrates and ubiquitously exists across all life forms [108]. CheR catalyzes the methylation of specific glutamyl residues in chemoreceptors and, together with phosphorylated CheB, regulates the methylation level of the chemoreceptors to adapt the new signal strength. Like CheB, different CheRs have different specificities and affinities to tether to the C-terminal pentapeptide of the chemoreceptor and to methylate the chemoreceptors [105,109,110]. The structure of CheR has been solved in some bacteria. For example, CheRs from *S. typhimurium* and *B. subtilis* possess a similar structure with a smaller N-terminal helical domain linked to a larger C-terminal α/β domain via a single polypeptide linker and a small antiparallel β sheet subdomain appended to the C-terminal domain [108]. Sequence comparison and structure prediction showed that *A. fabrum* CheR is highly similar to these experimentally studied CheRs. Therefore, *A. fabrum* CheR should have functions highly similar to other homologous CheRs, although no experimental study on *A. fabrum* CheR is reported.

CheD homologs are found in most nonenteric chemotactic bacteria, although they are not present in enterobacteria like *E. coli*. The best-studied CheD is *B. subtilis* CheD [111,112]. Other relatively well-characterized CheDs are the CheDs from *Thermotoga maritima* [113], *Borrelia burgdorferi* [114], and *Pseudomonas aeruginosa* [115]. CheD not only deamidates the glutamine residues in a conserved structural motif of the chemoreceptors to create some methyl-accepting sites, but also hydrolyzes glutamyl-methylesters at the modification sites of the chemoreceptors [113]. However, all these experimentally studied CheDs possess specificity to the chemoreceptor. They can only bind and deamidate some specific chemoreceptors. The *A*. *fabrum atu0521* gene was annotated to encode CheD. *A. fabrum* CheD shows relatively low sequence identity to other CheD homologs (14.36% identity with *B. subtilis* CheD, 29.12% identity with *T. maritima* CheD, 26.5% identity with *P. aeruginosa* CheD, 22.16% identity with *B. burgdorferi* CheD). Therefore, more experimental data are required to verify the functions of *A. fabrum* CheD.

### 3.6. The Other Chemosensory-Related Proteins Encoded by the A. faberum Genome

Besides the above-mentioned chemotaxis proteins, the chemotaxis gene cluster in *A. fabrum* has two genes, *atu0515* and *atu0522*, which are annotated to encode conserved hypothetical proteins, respectively. According to the locations of these two genes in the gene cluster and the sequence analysis of their encoded proteins, *atu0515*-encoded proteins and *atu0522*-encoded proteins are, respectively, very similar to the CheS and CheT that were identified in *Sinorhizobium meliloti*.

The *atu0515*-encoded protein has a sequence highly homologous to *S. meliloti* CheS, whose homolog is not present in the model chemotaxis system, but is found in other members of α-proteobacteria, such as *Sinorhizobium medicae*, *A. fabrum*, *Rhizobium leguminosarum*, *Rhodobacter sphaeroides*, and *Caulobacter crescentus* [47]. CheS is a single-domain protein containing the STAS (sulfate transporter and anti-sigma factor antagonist) domain. Most of the small STAS domain-only proteins are anti-sigma factor antagonists, which are involved in the regulation of sigma factors and anti-sigma factors [116]. *S. meliloti* CheS can bind with CheA to form a CheA/CheS complex and affect the binding of CheA to CheY. Furthermore, *S. meliloti* CheS helps CheA to differentiate two CheYs and results in the transfer of the phosphoryl group from one CheY to another CheY via CheA [47]. *A. fabrum* CheS exhibits a sequence identity of 58.59% to *S. meliloti* CheS, demonstrating that *A. fabrum* CheS should have functions similar to *S. meliloti* CheS.

The protein encoded by the *atu0522* gene is highly homologous to the most recently reported CheT that was identified in *S. meliloti* [44]. The genes encoding CheT homologs are also found in other closely related α-proteobacteria such as *S. medicae*, *R. leguminosarum*, *C. crescentus*, and *Hoeflea* sp. 10. The sequences of CheT homologs do not show significant homology with any known chemotactic protein in the chemotaxis paradigm model. Sequence analysis shows that CheT has a phosphatase motif (DXXXQ sequence). CheT structure predicted by AlphaFold is highly similar to the crystal structure of *E. coli* CheZ, which is a CheY-specific phosphatase terminating the CheY response signal in the paradigm model. *S. meliloti* CheT can differentiate two CheYs and enhance the dephosphorylation of only one phosphorylated CheY. *S. meliloti* CheT can form a complex with CheR and both the CheT and CheR in the CheT/CheR complex can maintain their original activities [44]. The protein encoded by the *A. fabrum atu0522* gene exhibits a sequence identity of 70% to *S. meliloti* CheT. Therefore, the functions of *atu0522*-encoded CheT should be very similar to those of *S. meliloti* CheT.

## 4. The Possible Signaling Pathways in the *Agrobacterium fabrum* Chemosensory System

The chemosensory signaling system is particularly important to phytopathogenic bacteria [117,118,119]. Up to 90% of phytopathogenic bacteria have chemotaxis-related genes [120], whereas such genes are found in only 47% of all bacteria [79]. Phytopathogenic bacteria often have multiple chemosensory systems and a high number of chemoreceptors, almost twice the number in those species not classified as plant-associated bacteria [120]. However, the *A. fabrum* genome encodes only one histidine kinase CheA although it contains twenty chemoreceptor-encoding genes, two CheW-encoding genes, and two CheY-encoding genes. CheA is the central component of the chemosensory signaling system. Based on the understanding of the established chemosensory system, *A. fabrum* can only assemble one complete set of the chemosensory system due to the presence of only one histidine kinase CheA, but one chemosensory system cannot illustrate why *A. fabrum* needs two CheWs and two CheYs and how *A. fabrum* possibly utilizes one chemosensory system to regulate various other physiological processes except for the flagellar motility. *A. fabrum* also has inducible type IV pili [34]. However, it remains unknown whether and how the chemosensory system mediates type IV pilus-based motility. Moreover, the *A. fabrum* genome encodes three additional chemosensory-related proteins (CheD, CheS, and CheT) that are not found in the prototypical model chemosensory system. The molecular mechanism of signal transduction and regulation unveiled in the model chemosensory system is also unable to rationally illustrate why the *A. fabrum* chemosensory signaling system requires CheS, CheT, and CheD. Therefore, the *A. fabrum* chemosensory signaling system should be more complicated than the established chemosensory systems.

It should be pointed out that the characteristic composition of *A. fabrum* chemosensory-related protein components is not unique in bacteria. As mentioned above, many bacteria, such as *Azorhizobium caulinodans* [46,97], *Rhizobium meliloti* [45], *Sinorhizobium medicae*, *Rhizobium leguminosarum*, *Rhodobacter sphaeroides*, and *Caulobacter crescentus* [47], also have the composition of one histidine kinase for two coupling proteins and multiple response regulators, as well as the additional proteins (CheD, CheS, and CheT) that are not found in the prototypical chemosensory system. Therefore, the molecular mechanism of signal transduction and regulation in the *A. fabrum* chemosensory system may become a typical representative of a large group of plant-associated bacteria.

By integrating all published experimental data on the *A. fabrum* chemosensory signaling system and comparing its chemosensory protein components with homologs, we propose a model to describe two signaling pathways in the chemosensory system of *A. fabrum* and the possible regulation between the two pathways (Figure 3). In this model, since two CheWs have different affinities to couple CheA to different chemoreceptors [93], each CheW can preferentially couple CheA to the chemoreceptors that have high affinities to the corresponding CheW and form the MCP–CheW–CheA CSU without another CheW. As a result, two CheWs are possibly able to assemble two types of CSUs: the MCP–CheW_2617_–CheA CSU and the MCP–CheW_2075_–CheA CSU. One CheW can be partially complemented by the other only when the molecular concentration of one CheW is too small to compete with the other. These two CSUs may have distinct affinities to bind two CheYs due to the different CheWs [100] and accordingly each CSU can only phosphorylate the CheY possessing high affinity. According to the effects of two CheWs and two CheYs on *A. fabrum* chemotaxis [93,100], the CSU with CheW_2617_ may preferentially phosphorylate CheY_520_, whereas the CSU with CheW_2075_ may mainly phosphorylate CheY_516_. Two *A. fabrum* CheYs may be used to regulate different physiological functions. Only CheY_520_ was supposed to directly regulate flagellar motor. Under normal conditions, each CSU phosphorylates its cognate CheY to regulate specific downstream functions. Consequently, the *A. fabrum* chemosensory system may form two signaling pathways (Figure 3). Cross-phosphorylation may occur only when the cognate CheY is limiting.

CheS in *S. meliloti* (a species closely related to *A. fabrum*) assists CheA in distinguishing between the two CheYs, enabling CheA to mediate cyclic phosphoryl group transfer between them via a CheA–relay mechanism [47]. CheT directly mediates the differential dephosphorylation of the two phosphorylated CheYs [44]. Through these mechanisms, we speculate that CheS and CheT collectively regulate the transfer and distribution of phosphoryl groups between the two CheYs. Therefore, it is plausible that *A. fabrum* CheS and CheT modulate signal crosstalk between the two pathways by regulating the CheA-dependent phosphoryl group shuttling between the two CheYs via CheA and the CheT-mediated differential dephosphorylation of the two phosphorylated CheYs.

CheD is hypothesized to regulate the methyl-accepting sites in the chemoreceptor. As in the prototypical chemotaxis system, CheR and phosphorylated CheB modulate chemoreceptor sensitivity in response to changing signal strengths through methylation/demethylation. However, once bacteria find a favorable growth environment through chemotaxis, they must also adjust their metabolism to adapt to the new conditions. For long-term adaptation to a new environment, bacteria need to regulate the expression of certain genes [121]. In addition, *A. fabrum* CheS is a small protein composed solely of a STAS domain and is encoded by a gene within the chemotaxis gene cluster. Since most small STAS domain-only proteins function as anti-sigma factor antagonists, which regulate gene expression, this model proposes that CheS might also regulate gene expression through interaction with sigma factors.

## 5. Perspectives on Multiple Chemosensory Pathways Sharing a Histidine Kinase

In summary, the chemosensory systems of bacteria exhibit diversity due to differences in their ecological living conditions. Some bacteria, such as *E. coli*, inhabit relatively stable environments; their chemosensory systems are correspondingly simple, consisting of only a few chemoreceptors, one CheW, one CheA, and one CheY, and these systems solely regulate chemotaxis. In contrast, bacteria living in more dynamic environments—*P. aeruginosa*, for instance—possess highly intricate chemosensory systems. Not only have they evolved more chemoreceptors, but they also feature multiple chemosensory systems and use different systems to regulate a variety of physiological functions beyond chemotaxis. However, a group of plant-associated bacteria, represented by *A. fabrum*, employs a strategy involving more chemoreceptors, one CheA for two (or more) CheWs and two (or more) CheYs. We may call this the “one-system two-pathways” strategy. This allows them to respond to diverse signals in various ecological environments and fulfill the need to regulate multiple physiological functions.

In the chemosensory system of *A. fabrum*, the two CheWs may couple CheA to different chemoreceptors based on their distinct affinities for different chemoreceptors, forming two types of CSUs. These two CSUs may endow CheA with different affinities for the two CheYs due to differences in the CheWs, thereby enabling discrimination between the two CheYs and regulating their phosphorylation levels, respectively. The two phosphorylated CheYs may independently regulate different physiological functions. Therefore, a single histidine kinase may orchestrate two signaling pathways. However, experimental validation of the model proposed in Figure 3 requires addressing the following key questions:How is the chemosensory array assembled in *A. fabrum*?

It remains unresolved whether the chemosensory supramolecular array is assembled via the initial formation of a chemoreceptor array with CheW later coupling CheA, or via pre-assembled CSUs that pack into the array. It is also unknown whether all 20 chemoreceptors form a single supramolecular array through random combination or assemble into relatively independent multiple supramolecular arrays based on differences in the signals they recognize or their structures.

Given the potential diversity of the 20 chemoreceptors, the assembly process of the chemosensory array is inherently complex. First, since the 20 chemoreceptor-encoding genes are controlled by different operators, some of them may be regulated by *A. fabrum*’s life cycle and growth niches, making it impossible for all 20 chemoreceptors to be simultaneously and stably expressed. Second, the six cytoplasmic chemoreceptors may form distinct arrays from transmembrane ones. Given the dynamic regulation and spatial distribution of the chemoreceptors, they are unlikely to form identical molecular arrays across different cells or even at different stages within the same cell.

2.How are the signals transduced in the *A. fabrum* chemosensory system?

If chemoreceptors recognizing different signals form separate arrays, the two pathways described in Figure 3 would possibly exist in reality. Different signals are basically transduced independently of each other, and signal exchange may occur between different arrays. In this case, CheS and CheT would function as the signal coordinators (or mediators) of the two pathways via shuttling among arrays.

In contrast, if all 20 chemoreceptors assemble into a single array, signals from different chemoreceptors may converge on CheA, which then distributes phosphate groups to different regulators to coordinate distinct physiological functions. In such a scenario, CheS and CheT might localize to the array and function to assist CheA in distributing phosphate groups to different regulators.

It remains unclear whether *A. fabrum* possesses additional CheA-phosphorylatable response regulators beyond CheB, CheY1, and CheY2 and whether the two CheW proteins influence CheA’s affinities for different response regulators, thereby influencing CheA’s ability to distribute phosphate groups and coordinate signal transduction. Furthermore, no conclusive evidence currently demonstrates whether CheY2 or other potential CheA-phosphorylatable regulators regulate non-chemotactic physiological functions.

3.Does the *A. fabrum* chemosensory system play a role in long-term adaptation?

Currently, it remains unclear whether the *A. fabrum* chemosensory system contributes to the long-term adaptation process. This is because we lack experimental evidence demonstrating that CheS interacts with any sigma factor or anti-sigma factor. Therefore, it is still uncertain whether *A. fabrum* CheS can regulate gene expression.

## Figures and Tables

**Figure 1 microorganisms-13-02556-f001:**
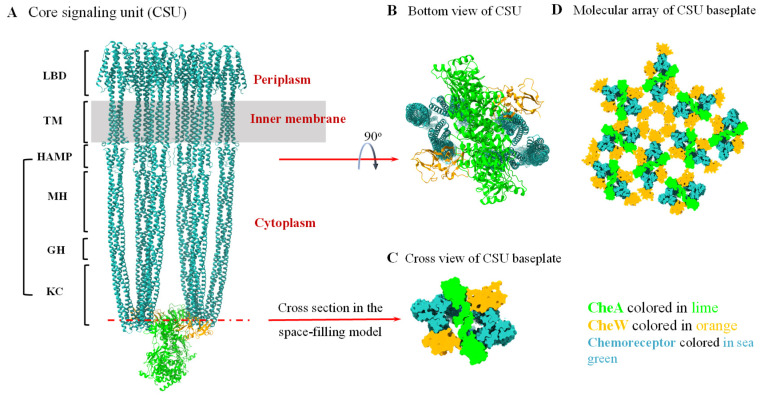
Structure of core signaling unit (CSU) and architecture of the chemosensory array baseplate in the prototypical chemosensory system of *E. coli*. (**A**) All-atom ribbon model of the native *E. coli* CSU (PDB 8C5V). Specific regions of the chemoreceptors are labeled in the left side. LBD: the ligand-binding domain; TM: transmembrane helical region; CSD: cytoplasmic signaling domain; HAMP: domain existing in histidine kinases, adenylate cyclases, methyl-accepting chemotaxis proteins, and phosphatases; MH: methylation helix bundle; GH: glycine hinge; KC: kinase control region. (**B**) The bottom view of the CSU showing the baseplate structure of the CSU. (**C**) Cross sectional view of the CSU baseplate in the space-filling CSU model showing the roughly hexagonal baseplate. (**D**) The chemosensory array baseplate model assembled from the cross-sectional views of the repeated CSU baseplate. CheW and CheA P5-domain form a hexameric ring in an alternating order. Additional CheW molecules may change the ratio of CheA P5-domain to CheW molecules in the array and possibly form a hexameric CheW ring without CheA P5-domain.

**Figure 2 microorganisms-13-02556-f002:**
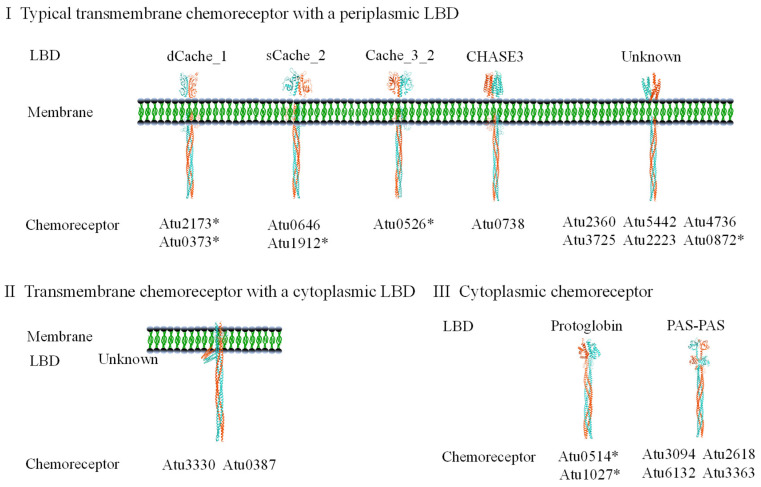
The chemoreceptors annotated in the *A. fabrum* C58 genome. Chemoreceptors are named after the genes that encode them. Chemoreceptors marked with an asterisk (*) has been experimentally characterized. LBDs were predicted based on Pfam database annotations [83]. Categories were assigned based on predicted transmembrane region number and localization in each chemoreceptor.

**Figure 3 microorganisms-13-02556-f003:**
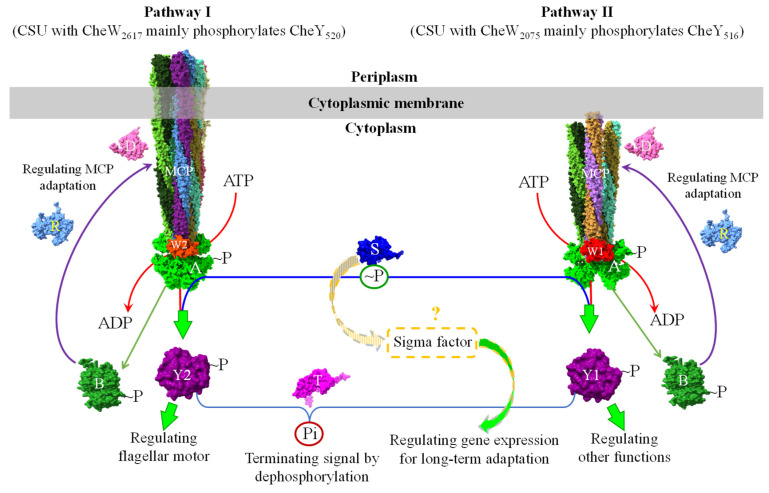
A schematic model describing the chemosensory signaling system of *A. fabrum*. Two CheWs couple CheA to different chemoreceptors depending on their affinities to the chemoreceptors to assemble two different CSUs (core signaling units) and thus possibly form two chemosensory signaling pathways, Pathway I and Pathway II. Two CheWs may affect the affinities of CheA to two CheYs. CSU with CheW_2617_ mainly (or preferentially) phosphorylates CheY_520_. CSU with CheW_2075_ mainly (or preferentially) phosphorylates CheY_516_. CheY_520_ mainly regulates flagellar motor, whereas CheY_516_ is supposed to mainly regulate other functions. CheS is supposed to regulate the distribution of the phosphoryl group between two CheYs via CheA and the gene expression via interacting with some sigma factors. CheT possesses different capabilities to dephosphorylate two phosphorylated CheYs. Thus, CheS and CheT possibly mediate the signal interchange between two pathways. CheD, CheR and the phosphorylated CheB regulate the chemoreceptors’ sensitivities to adapt the changing ligand concentration. For the detailed functions of these proteins, please see the text. Abbreviations: A, CheA; B, CheB; CSU, core signaling unit; D, CheD; MCP, chemoreceptor; R, CheR; S, CheS (encoded by *atu0515*); T, CheT (encoded by *atu0522*); W1, CheW1 (encoded by *atu2075*; CheW_2075_); W2, CheW2 (encoded by *atu2617*; CheW_2617_); Y1, CheY1 (encoded by *atu0516*; CheY_516_); Y2, CheY2 (encoded by *atu0520*; CheY_520_); ~P, phosphoryl group; Pi, inorganic phosphate.

## Data Availability

Data derived from public domain resources. The data presented in this study are available in [MiST4.0: Microbial Signal Transduction Da-241 tabase (mistdb.com)] at [https://mistdb.com/mist/genomes/GCF_000092025.1/signal-genes?ranks=chemotaxis] accessed on 25 August 2025, reference number (reference [83]). These data were derived from the following resources available in the public domain: [https://alphafold.ebi.ac.uk/], accessed on 25 August 2025.

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
