# Peer review of "Agrobacterium fabrum (tumefaciens) Chemosensory System: A Typical Model of One Histidine Kinase for Two Coupling Proteins and Multiple Response Regulators"

_microorganisms, 2025, doi:10.3390/microorganisms13112556_

Round 1
Reviewer 1 Report
Comments and Suggestions for Authors
“Agrobacterium fabrum (tumefaciens) Chemosensory System: A Typical Model of One Histidine Kinase for Two Coupling Proteins and Multiple Response Regulators”
The manuscript under review presents a comprehensive and timely survey of the chemosensory machinery of Agrobacterium fabrum, a system that diverges in several essential ways from the canonical paradigm established in Escherichia coli. The authors begin with a concise yet informative introduction to the general principles of bacterial chemotaxis, and then progressively narrow their focus toward the distinctive features of A. fabrum, which encodes a single CheA histidine kinase but combines it with multiple coupling proteins and response regulators. On this basis, they propose a model in which two partially independent signalling pathways can coexist, both relying on the same central kinase.
The article succeeds in highlighting why A. Fabrum deserves special attention. This bacterium is not only a historically significant plant pathogen and biotechnological tool but also an intriguing example of signal network diversification in α-proteobacteria. The review is carefully structured: after outlining the evolutionary and structural background of chemotaxis systems, the authors discuss in detail the genetic organisation of the relevant loci, the properties of chemoreceptors, and the roles of less conventional components, such as CheS and CheT. Figures, particularly the schematics illustrating the supramolecular architecture and the proposed two-pathway model, are well-designed and help readers visualise a complex system.
There are, however, some aspects where the manuscript would benefit from refinement. At several points, the text moves from established knowledge to speculative interpretation without a sufficiently sharp distinction. For example, the suggestion that CheS might influence gene expression through sigma-factor interactions, or that CheT differentially dephosphorylates the two CheY variants, is certainly intriguing, but remains conjectural in the absence of direct evidence in A. fabrum. Such hypotheses deserve mention, but they should be presented with clearer caveats to avoid giving an impression of experimental confirmation where none exists. Another point concerns the emphasis on the two-pathway model. While it is reasonable and consistent with comparative data from related organisms, the possibility of alternative arrangements—such as heterogeneous arrays of chemoreceptors feeding into a single integrated signalling unit—receives little attention. A brief discussion of competing scenarios would enrich the review and underscore the provisional nature of the proposed scheme. From a presentational perspective, some information could be organised more effectively. The section on methyl-accepting chemotaxis proteins, for instance, is dense with detail. A summary table listing all 20 MCPs, their predicted ligand-binding domains, membrane topology, and presence or absence of the C-terminal pentapeptide would greatly aid readers. Likewise, consistency in terminology would be welcome. The manuscript alternates between “chemotaxis system,” “chemosensory system,” and “signalling pathway” in ways that might confuse a non-specialist audience; settling on a single preferred term would improve clarity.
These issues aside, the manuscript demonstrates a clear command of the literature, with citations that extend to the most recent studies in the field. It also makes a valuable contribution by synthesising disparate lines of evidence into a coherent model that can guide future work. The speculative elements, rather than being flaws per se, highlight precisely where experimentalists should direct their efforts in the coming years. In conclusion, this review is a significant and well-executed piece of scholarship that will interest researchers studying bacterial chemotaxis, plant–microbe interactions and signalling diversity. With minor revisions, chiefly clarifying which statements are hypotheses, providing a more balanced discussion of alternative models, and enhancing the presentation of MCP data—the manuscript will be ready for publication. My recommendation is acceptance after minor revision. Good luck.
Reviewer 2 Report
Comments and Suggestions for Authors
Title
Agrobacterium fabrum (tumefaciens) Chemosensory System: A Typical Model of One Histidine Kinase for Two Coupling Proteins and Multiple Response Regulators
Abstract
Line 11: What type of signals?
Line 20: What is the prototypical system based on? Or what makes the E. coli system different from that of A. fabrum?
Lines 21-24: What is the purpose of comparing the different motility mechanisms?
Line 24: What other species?
Line 28: It would outweigh the knowledge that these were compiled, analyzed, and discussed with other studies.
Overall, the summary is well developed; however, the differences with other similar reviews should be mentioned. Highlight the novelty of the study compared to those already available.
Keywords
Do not repeat words from the title, which are required for indexing.
- Introduction
Citations should be numbered, according to the number of appearances in the text.
Line 55: Avoid subjective words like good (completely, entirely, etc.).
Lines 53-62: This information can also be found in other reviews.
Line 79: Is this amount sufficient/scarce?
Lines 79-81: What does this information entail?
The introduction contains very general information that does not reflect the importance of conducting the corresponding review. It could be based on knowledge gaps on these topics and how this compilation attempts to fill them.
- Prototype of chemosensory signaling system and coexistence of multiple systems
Lines 85-88: This statement should be supported by a citation.
Lines 100-103: Why are these groups not explained in detail later?
In section 2: Prototype of chemosensory signaling system and coexistence of multiple systems (lines 85-108), an introductory section on the topic of coexistence of multiple systems is not mentioned, but is later explained in detail in section 2.2.
Section 2.2 Coexistence of multiple chemosensory systems, a schematic could be adapted to better understand this system. For example, the previous schematic and add the differences that exist with the coexistence of multiple systems.
- Chemosensory-related proteins encoded by Agrobacterium fabrum genome
It should be delineated whether A. fabrum has a prototype system (2.1) or a coexisting one (2.2).
Line 207: If only C58 is mentioned, only reference 116 mentions that the reported study was conducted on this strain. Were the other studies cited on this strain?
Lines 215-222: These statements should be supported with citations.
Lines 230-236: Which author describes this process?
Line 240: What is known about the other two chemoreceptors? This is where a review could fill in the gaps in knowledge, as long as information on the topic exists.
Lines 266-274 do not contain citations.
Lines 297-306 contain several passages throughout the text that are statements. Sometimes, a cellular regulatory process is described, while others provide figures and data. I was wondering if these are your own results, or why isn't a citation included?
There are sections, such as 3.2, that are very short. The purpose of a review is to exhaustively search for information, organize it, and interpret it, which is lacking in many sections.
- The possible signaling pathways in Agrobacterium fabrum chemosensory system
Lines 471-473: What would this entail?
Figure 3. It strikes me that, using only data reported by other authors, they were able to construct a model describing the chemosensory signaling system of A. fabrum, without any in vitro or in silico experimental work?
- Perspectives on multiple chemosensory pathways sharing a histidine kinase
A conclusion is needed
Are the existing doubts not resolved by the generated model?
How would the authors answer these questions based on the entire review they conducted?
It is suggested that they support their model with information that has only been validated experimentally or in silico, and that they clarify that all information was taken only from work on the C58 strain.
Reviewer 3 Report
Comments and Suggestions for Authors
I checked this manuscript and described comments below.
This paper is an excellent review of histidine kinase from Agrobacterium fabrum (tumefaciens).
The content is correct, but it is difficult to follow the protein symbols from the text, so it would be better if there was a table of protein symbols and protein names.
Regarding "3.3. Histidine kinase, CheA," there is a description of the protein domains, but I think it would be better to illustrate the domain structure.
I don't think this paper has major problems and grammatical problems.
Round 2
Reviewer 2 Report
Comments and Suggestions for Authors
Title
Agrobacterium fabrum (tumefaciens) Chemosensory System: A Typical Model of One Histidine Kinase for Two Coupling Proteins and Multiple Response Regulators
Abstract
Substantially improved based on feedback.
Keywords
Repetition of those in the title was avoided.
- Introduction
Substantially improved; the manuscript style was adapted to the journal's guidelines.
- Prototype of chemosensory signaling system and coexistence of multiple systems
The corresponding clarifications were made.